# Responses of Arbuscular Mycorrhizal Fungi and Plant Communities to Long-Term Mining and Passive Restoration

**DOI:** 10.3390/plants14040580

**Published:** 2025-02-14

**Authors:** Sofía Yasmín Utge Perri, María Victoria Valerga Fernández, Adalgisa Scotti, Roxana Paula Colombo, Florencia González, Lautaro Valenzuela, Alicia Margarita Godeas, Vanesa Analía Silvani

**Affiliations:** 1Instituto de Biodiversidad y Biología Experimental y Aplicada, CONICET-UBA, Facultad de Ciencias Exactas y Naturales, Universidad de Buenos Aires, Buenos Aires C1428EGA, Argentina; sofiautge@gmail.com (S.Y.U.P.); colomboroxanap@gmail.com (R.P.C.); godeas@bg.fcen.uba.ar (A.M.G.); 2Departamento de Biodiversidad y Biología Experimental, Facultad de Ciencias Exactas y Naturales, Universidad de Buenos Aires, Buenos Aires C1428EGA, Argentina; vickyvalfer@gmail.com; 3Laboratorio Bioambiental, Comisión Nacional de Energía Atómica, International Center for Earth Sciences, San Rafael M5600, Argentina; adalgisascotti@gmail.com; 4Facultad de Ciencias Exactas y Naturales, Universidad Nacional de Cuyo, Mendoza M5500, Argentina; 5División Metodologías Nucleares de Análisis, Gerencia Química, Centro Atómico Constituyentes, Comisión Nacional de Energía Atómica, Buenos Aires B1650LWP, Argentina; florenciagonzalez@cnea.gob.ar (F.G.); frxcnea@gmail.com (L.V.)

**Keywords:** mining activities, arbuscular mycorrhizal fungi, heavy metal, life history strategies

## Abstract

Mining activities cause strong soil alterations, such as heavy metal (HM) pollution, which decreases the diversity of plant communities and rhizospheric microorganisms, including arbuscular mycorrhizal (AM) fungi. The polymetallic Paramillos de Uspallata mine in the Andes Mountains, the first mining exploitation in Argentina, provides a unique scenario to study AM fungal resilience after long-term disturbance following over 40 years of inactivity. This study aimed to analyze mycorrhizal status and AM fungal communities in the mine and a nearby unexploited area and to evaluate their associations with soil parameters to elucidate life history strategies. Long-term exposure to elevated Fe, Pb, Zn, and Ag concentrations and high electrical conductivity (EC, 5.46 mS/cm) led to the dominance of *Entrophospora infrequens* in association with *Pappostipa speciosa*, demonstrating that this AM species is a stress-tolerant strategist in symbiosis with a pioneer perennial plant, resilient in the most impacted mine areas. In contrast, the unexploited area, with an EC of 0.48 mS/cm and low HM contents, supported competitive and ruderal species, revealing distinct ecological strategies of AM fungi in disturbed versus undisturbed environments. These findings highlight the potential of *E. infrequens* for bioremediation and ecological restoration in post-mining landscapes.

## 1. Introduction

Arbuscular mycorrhizae are mutualistic symbiotic associations established between most terrestrial plants and certain soil fungi of the Phylum Glomeromycota [1]. In this symbiosis, the arbuscular mycorrhizal (AM) fungus provides nutrients and water to its host plant in exchange for photoassimilates, which it uses to complete its life cycle [2]. Additionally, AM fungi protect their host plant from biotic and abiotic stress, including heavy metal (HM) soil pollution [3,4]. It is well known that AM fungi mitigate HM toxicity by uptaking and sequestering them in their structures as a mechanism of HM alleviation [5,6]. Furthermore, these fungi contribute to soil quality by stabilizing and forming aggregates. They also play key roles in ecosystem function by facilitating productivity and diversity of plant communities, as well as re-establishing rhizospheric microbial communities in degraded environments [7,8]. Therefore, AM fungi perform important functions in polluted and degraded sites, such as abandoned mines [9,10], where certain species are frequently associated with adapted or tolerant plant species under stressful conditions [11,12,13,14].

While most studies in mine sites have explored the impact on plants and AM fungal communities, the ecological role of AM fungal species in those habitats remains poorly understood [15]. Ecological conceptual frameworks aim to provide predictive tools to understand the ecological function of AM symbiosis [16], particularly in disturbed ecosystems, where AM fungi must adapt to extreme environmental pressures. Based on the prevailing environmental conditions (disturbance or stress), AM fungi are classified into three life history strategies: competitors, stress-tolerant, and ruderals, as described by Grime’s C–S–R model [17,18]. Competitors, in low-stress and low-disturbance environments, delay reproduction and invest in structures that optimize resource acquisition. By contrast, stress-tolerant fungi endure in suboptimal conditions through strategies that conserve resources. Finally, ruderals cope with frequent disturbance by rapidly colonizing niches, reproducing quickly, and producing low-cost biomass. Although many studies have applied life history strategies in AM fungi, they often focus on a limited subset of easily cultivable AM fungal species, most of them isolated from temperate soils [18,19,20]. Therefore, it is important to investigate other environmental conditions in order to gather information on additional AM species, enabling their classification based on the life history strategies.

Mining activities profoundly impact the biodiversity and functionality of plants and their microbial communities by altering the physical and chemical properties of soils [15]. Among the most severe environmental consequences, with long-term ecological risks, is HM pollution of soils [11,21,22]. The Andes Mountains in South America have a long mining history, with the Paramillos de Uspallata mine, located in the Andean foothills, being the oldest in Argentina. This mine was exploited for silver (Ag), gold (Au), lead (Pb), zinc (Zn), and copper (Cu) from pre-Hispanic times until the late 19th century and has remained abandoned since the early 1980s [23]. Currently, the mine exhibits different disturbed areas, including a waste dump, building ruins, and remnants of mineral processing structures, where passive natural restoration has been ongoing for over four decades. However, to our knowledge, the long-term effects of mining on vegetation and AM fungi, as well as the processes of natural passive restoration in this region, remain largely unexplored. Previous studies have highlighted that AM fungal diversity in Andean mining-affected soils is still poorly understood [15,24]. For instance, Cornejo et al. [3] evaluated the role of AM fungi in the recovery of Cu-contaminated mining soils in Chile, while Suarez et al. [25] studied AM fungal communities in artisanal and small-scale gold mining sites in southeastern Ecuador. In the latter study, no specialized AM species were detected in HM-contaminated sites; instead, generalist species from Glomeraceae, adapted to a wide variety of habitats, were predominant. However, the environmental conditions of those sites differ markedly from those of the Paramillos mine. Notably, no studies have yet been performed on AM fungal communities associated with mining sites in Argentina, underscoring the need for further research in this region. Furthermore, an important question arising from this study is whether the different mining activities at Paramillos mine have led to distinct environmental recovery scenarios. If so, this site could serve as a model for studying ecological succession under natural passive restoration, providing valuable insights into long-term recovery processes in post-mining landscapes.

García de León et al. [26] provided field evidence that AM fungi are effective colonizers of disturbed sites and can shape plant community composition and diversity during succession. In this context, the study of the life history strategies of AM fungi is relevant to understanding their ecological roles and adaptive capacities, as these strategies may influence fungal responses to environmental factors and their contributions to ecosystem processes. Studies carried out in mining sites have shown that ruderal species (Glomeraceae) and stress-tolerant species (Acaulosporaceae) tend to dominate in these environments [11,25,27]. Likewise, an increase in the similarity of the mycorrhizal communities between rehabilitated and reference areas over time in a mining area was observed [28]. These findings suggest that AM fungal communities in abandoned mines may follow a predictable successional trajectory. In highly disturbed areas, stress-tolerant and ruderal species are likely to dominate initially, facilitating passive restoration by enhancing soil conditions and promoting plant colonization. As soil quality gradually improves, competitive AM fungi may emerge, supporting the establishment of more diverse plant and fungal communities. This transition could ultimately enhance ecosystem stability and resilience, contributing to the long-term ecological restoration of Paramillos de Uspallata mine.

Here, we hypothesize that the intensity and type of mining activities at the Paramillos de Uspallata mine influence the differential recovery of soil properties, vegetation, and mycorrhizal communities by impacting soil degradation and exerting selective pressures on biotic components. The AM fungal communities in this abandoned mine may contribute to mitigating HM toxicity and facilitate passive ecological restoration through stress-tolerant and ruderal life history strategies. To test this hypothesis, we analyzed the mycorrhizal status of vegetation and the abundance and diversity of AM communities within the mine and in a nearby unexploited area, and we examined their associations with environmental parameters to elucidate their life history strategies. The findings of this study will enhance our understanding of the ecological roles and adaptive mechanisms of AM fungi in ecosystem recovery under prolonged mining-induced stress. Moreover, this research will contribute to the development of restoration strategies for mining-affected regions, emphasizing the role of AM fungi in promoting ecosystem resilience.

## 2. Results

### 2.1. Soil Properties After Mining and Passive Restoration

The physicochemical properties and chemical element concentrations of soil and substrate from the mine areas and an undisturbed area outside the mine (Off-mine area) are shown in Table 1. Soils from the Urban Ruins area, Explotation area 1 (Exp 1), and Off-mine area did not differ statistically and were moderately alkaline (approx. pH = 8.3), whereas soil from the Explotation area 2 (Exp 2) area was slightly alkaline (pH = 7.8 ± 0.4). In contrast, the substrate from the dump at the Exp 2 area exhibited an acidic pH value (5.5 ± 0.1). Electric Conductivity (EC) and Total Dissolved Solutes (TDS) values were highest in Exp 2 (5.46 ± 2.5 mS cm^−1^ and 1.49 ± 0.4 ppt, respectively) and lowest in the unexploited area (0.48 ± 0.2 mS cm^−1^ and 0.21 ± 0.1 ppt, respectively). Exp 2 area also recorded the lowest Ca and Mg concentrations (2.10 ± 0.10% and 0.78 ± 0.08%, respectively), while the dump substrate had low Na and K content (0.27 ± 0.03% and 0.5 ± 0.04%, respectively), but a significant elevated S content (8.40 ± 0.40%). Total Carbon (TC) and Total Phosphorus (TP) concentrations were low across most areas, with TC ranging from 5.47 to 7.10% and TP from 0.12 to 0.18%, except in the Urban Ruins area, where TP concentration slightly increased (0.43 ± 0.04%).

HM pollution was notably higher in Exp 2, with elevated concentrations of Fe, Pb, Zn, and Ag compared with the Off-mine area. Ag ions were not detected in soil samples from the non-impacted area, and sampling area Exp 2 had the highest Cu concentration (726 ± 68 mg kg^−1^), far exceeding levels in the Off-mine area (51 ± 10 mg kg^−1^) and other mine sites (182 to 248 mg kg^−1^). Cr levels were similar in the Urban Ruins and Exp 1 areas (70 ± 14 and 64 ± 13 mg kg^−1^, respectively), while V was exclusive to Exp 1 and the Off-mine area. Sb was detected in Exp 1 and Exp 2, with concentrations increasing from Exp 1 (180 ± 18 mg kg^−1^) to site J in Exp 2 (214 ± 21 mg kg^−1^). The non-impacted area exhibited the highest Sr concentration (301 ± 30 mg kg^−1^), whereas Exp 2 had the lowest (81 ± 16 mg kg^−1^). Ba levels ranged between 446 to 622 mg kg^−1^ across the mine and Off-mine areas, with the highest concentration in the dump substrate (1300 ± 100 mg kg^−1^). Cd was exclusively detected in Exp 2, while no Au ions were detected across all areas. Ni concentrations did not vary among sampling areas.

### 2.2. Plants and Mycorrhizal Status

At the sampling time, fourteen species (native or endemic) belonging to twelve plant families were identified within the Paramillos de Uspallata mine and its surrounding area, with a predominance of perennial herbs and subshrubs under vegetative state (Table 2). The most disturbed area (Exp 2) showed the lowest plant diversity, comprising only three families (Poaceae, Malvaceae, and Rosaceae) and four species, and a reduction in the functional type, as only hemicryptophytes and chamaephytes were present. Likewise, no plants were detected at the dump within Exp 2; therefore, this sampling site was excluded from mycorrhizal analyses. In contrast, the other study areas varied from five to six families and twelve plant species in total, encompassing all functional types. Some plant families were exclusively found in the Off-mine area (Geraniaceae and Violaceae), in Exp 1 (Cactaceae), and in Exp 2 (Rosaceae).

Only the AM type was detected within Paramillos de Uspallata mine and the unexploited area (Table 2, Appendix A), with the predominance of Arum-type colonization. Likewise, the Arum–Paris type was observed in the roots of some individuals of *Pappostipa speciosa*, *Junellia uniflora*, *Artemisia mendozana* var. *paramilloensis*, and *Tetraglochin alata* from different areas within and outside the mine (Table 2, Appendix A). In contrast, no root colonization was observed in two exotic therophytes, *Erodium cicutarium* and an unidentified species from the Brassicaceae family, nor in the sampled plants from the Boraginaceae, Malvaceae, and Geraniaceae families found within the mine and the unexploited area (Table 2, Figure 1).

The highest levels of AM root colonization were recorded in the Urban Ruins area, Exp 1, and the Off-mine area for most plant species (Figure 1). In Exp 2, *T. alata* plants exhibited high levels of AM root colonization, in contrast to other plant species in this disturbed area. Plants of *P. speciosa*, found in all areas within the mine and its surroundings, were colonized by AM fungi, with the exception of a few individuals in the most disturbed area (Exp 2). The plant species *Senecio uspallatensis* and *A. mendozana* var. *paramilloensis* (Asteraceae family), both included in the red list of threatened species (categories 4 and 5), exhibited medium to high levels of AM root colonization in the less disturbed area (Exp 1) and in the Off-mine area, respectively (Figure 1).

### 2.3. AM Fungal Communities

The AM spore density did not show statistically significant differences between the areas impacted by mining activity and the undisturbed area (F = 1.74; *p* = 0.22, Figure 2). However, great variability in spore number was observed among study areas, particularly in Exp 2. The spore density outside the mine and Exp 1 was 142 ± 56 and 164 ± 32 of AM spores per 10 g of dry soil, respectively. In contrast, the lowest values of spore density were found in the Urban Ruins area and Exp 2 (70 ± 12 and 99 ± 91, respectively). No AM fungal spores were found in the substrate of the dump at Exp 2.

In regards to the AM community composition and relative abundance of species (%), *Entrophospora etunicata* is the most abundant across all surveyed areas, except for Exp 2 (Figure 3). Its relative abundance was 26.8% in the Off-mine area, 27.4% in the Urban Ruins, and 24.8% in Exp 1. However, in Exp 2, the abundance of *E. etunicata* decreased substantially to 11.9%, while *Entrophospora infrequens* was the most abundant AM species, reaching 35% of the total observed spores, followed by *Acaulospora scrobiculata* that accounted for 18.3% of the spores (Figure 3). All spore characteristics typically associated with abiotic stress tolerance [29], such as small size, thick wall, dark-colored, and ornamented wall layers, were observed in *E. infrequens*.

The second most abundant AM species in the Urban Ruins area and Exp 1 was *Funneliformis geosporus,* with relative abundance values of 20.3 ± 6.7% and 13 ± 2.7%, respectively. In contrast, *Diversispora spurca*, *Funneliformis coronatus*, and *F. geosporus* were found in high proportions in the unexploited area (17.9%, 10.5%, and 10.1%, respectively). Also, the AM species *D. spurca* was abundant in both the urban area and Exp 1 but showed a marked decrease in Exp 2 (1.1%). 

A few species of *Rhizophagus* were found both inside and outside the mine, displaying relatively consistent proportions across different areas: 6.9% in the urban area, 9.4% in the Off-mine area and Exp 2, and 9.8% in Exp 1. The AM species *Glomus macrocarpum* was exclusively found within the mine, with a higher abundance in the urban area (8.1%) compared with the exploitation areas (3.1%). Likewise, *Acaulospora rehmii*, *Septoglomus constrictum,* and *Sacculospora baltica* were exclusively recovered from Exp 2, albeit in low proportions. In Exp 1, ornamented AM spores were observed but could not be assigned to any described species, suggesting the potential of a novel AM fungal species.

The proportionality of the AM fungal families in each area is shown in Figure 4. The Glomeraceae and Entrophosporaceae families were the most prevalent in all areas, although their proportions varied significantly in some of them. The Urban Ruins area and Exp 1 showed very similar percentages, with Glomeraceae representing 55.24% and 51.53%, respectively, followed by Entrophosporaceae at 36.19% and 34.83%, and Diversisporaceae at a lower proportion (8.6% in Urban Ruins and 11.6% in Exp 1). A similar proportion of the Entrophosporaceae family was assessed in the Off-mine area (34.33%), followed by Glomeraceae (39.79%).

The Off-mine area exhibited the highest AM diversity of AM families, represented by five families (Glomeraceae, Entrophosporaceae, Acaulosporaceae, Diversisporaceae, and Paraglomeraceae). This area showed a significant increase in Diversisporaceae species and a relative decline in Glomeraceae compared with the mine areas. In contrast, Exp 2 displayed a different pattern, with the Entrophosporaceae family more abundant (78.45%) and Glomeraceae contributing only 14.14%. In addition, Acaulosporaceae species were more abundant in Exp 2 (7.07%) compared with the non-exploited area (0.18%), and they were absent in the remaining mine areas. Paraglomeraceae species were exclusively detected in Exp 1 and outside the mine.

A tendency towards higher richness species (S) and diversity (D: Simpson index; H: Shannon–Wiener index) values was observed in the Off-mine area, Urban Ruins area, and Exp 1 compared with the most disturbed area, Exp 2 (Table 3). However, no statistically significant differences were found in these parameters (S: F = 2.15 and *p* = 0.16; H index: F = 1.1 and *p* = 0.39; D index: F = 1.10 and *p* = 0.39). The AM fungal communities in Exp 2 exhibited the highest D index value (0.44 ± 0.23), reflecting a lower diversity and a greater dominance of AM species compared with other areas. Particularly, at a sampling site within Exp 2, where mineral treatment pools were locateda high number of AM spores were observed, with *E. infrequens* dominating (191 out of 203 spores per 10 g of dry soil).

### 2.4. Relationships Between Soil Parameters and AM Communities

A Principal Component Analysis (PCA) was performed to evaluate the spatial distribution of physicochemical parameters of soils and mycorrhizal parameters (Figure 5). The first two components accounted for 86.9% of the total variance, with Principal Component 1 (PC1) and Principal Component 2 (PC2) explaining 66.3% and 20.6%, respectively. The results reveal clear differences among the areas, particularly for Exp 2, which was strongly positioned along the PC1 axis. The variables that contributed most to this separation included the physicochemical parameters such as EC and TDS of soils, as well as several HM (Sb, Zn, Fe, Cu, Mn, Pb, and Ag) and other chemical elements (S and K).

The Urban Ruins area exhibited a significant association with soil P content, whereas the Off-mine area was primarily characterized by AM fungal richness and the concentration of Sr and Mg, to a lesser extent. The Exp 1 area was associated with the density of AM spores and concentrations of V, Ba, and Na in soils. In contrast, Exp 2 was clearly separated from the other areas and was primarily associated with the HM content in soils (Pb, Ag, Mn, Zn, Fe, Sb, Cu), other chemical elements (K and S), and high TDS and EC values. In addition, AM fungal richness showed an inverse relationship with Exp 2, highlighting the strongly disturbed environment and the adverse effects of HM pollution on fungal and plant diversity.

The non-metric multidimensional scaling (NMDS) plot (Figure 6) based on the analysis of AM fungal species and soil parameters revealed an association between *A. scrobiculata*, *A. rehmii*, *E. infrequens*, *S. constrictum*, and *S. baltica* with the concentration of TDS, Zn, S, and EC values. Additionally, *R. intraradices* and *G. sinuosum* were linked to V content, while *Funneliformis* sp. and *R. microaggregatum* were associated with Ca, and *G. macrocarpum* correlated along the concentration of Cr.

Although the PERMANOVA results do not provide conclusive evidence for grouping species by the studied areas or environmental variables, a clear trend was observed, indicating a relationship between AM species and specific environmental soil factors.

The distance-based redundancy analysis (dbRDA) identified the soil parameter EC as a significant environmental variable (*p* = 0.04), explained by axis 1 with 81.3%, and closely linked to the abundance of the AM species *E. infrequens* in Exp 2 (Appendix A).

Considering the environmental conditions prevailing in each area, along with the results of PCA and NMDS, the life strategies (C-S-R) of each AM fungal species could be inferred. The distribution of AM functional groups (%) inside and outside the Paramillos de Uspallata mine is presented in Figure 7. The Urban Ruins and Exp 1 areas showed similar functional groups, with the ruderals species being the most abundant (55.2 and 53.6%, respectively), followed by stress-tolerant species (36.2 and 34.8%, respectively). In contrast, the Off-mine area exhibited more evenly proportions of each fungal strategy, with competitive species comprising 25.2%, ruderal species 40.3%, and 34.51% for stress-tolerant species. In the Exp 2 area, stress-tolerant AM fungi were the most abundant by far (85.5%), while ruderal species were present at lower proportions (14.1%), and competitive species were almost absent (0.34%).

## 3. Discussion

The abandoned Paramillos mine in Uspallata offers a unique environment shaped by a long-term mining history and over 40 years of natural recovery processes. Despite the prolonged period of inactivity in mining operations, soils in some areas of the mine exhibited elevated HM concentrations, with many of them (Zn, Ag, Pb, Cd, and Cu) exceeding ecotoxicity thresholds [30]. Nonetheless, certain plants and their associated AM fungal communities demonstrated the ability to thrive in this highly disturbed environment. Notably, in the Exp 2 area, encompassing mineral extraction and treatment activities, and the highest recorded levels of Ag, Cu, Mn, Fe, Pb, and Zn and EC value, four plant species and fifteen AM fungal species have successfully been established. However, no vegetation and AM propagules were observed in the mine dump within this area, where a low pH value and high sulfide content were recorded. Moreover, the oxidation of sulfide-rich waste rocks likely facilitated HM solubilization [31], creating extreme environmental conditions that further hindered plant establishment and disrupted AM fungal communities by probably affecting their survival and functionality [32]. In contrast, no HM pollution has been detected outside the mine area, demonstrating the impacts of mining activities on the environment.

Plants have evolved diverse strategies to thrive in arid and HM-polluted habitats, including the establishment of AM symbiosis [33,34]. In the areas under study, fourteen distinct plant species were found, ten of which established AM associations, showing the potential of AM symbiosis to overcome adverse conditions. To our knowledge, we reported for the first time that plant species *F. patagonica* (Solanaceae), *J. uniflora* (Verbenaceae), *T. alata* (Rosaceae), *M. glomerata* (Cactaceae), *S. uspallatensis*, and *A. mendozana* var. *paramilloensis* (Asteraceae) were colonized by AM fungi, expanding the global mycorrhizal status database. The discovery that endangered *S. uspallatensis* and *A. mendozana* species form this association should be attended to in restoration and conservation programs, given the benefits of AM symbiosis for vegetation establishment in arid and disturbed environments. The co-introduction of plant seeds and native AM fungi, along with other beneficial rhizosphere microbiota, can enhance the establishment and resilience of endangered species [35,36], as well as support revegetation programs through transplants of mycorrhizal plants from nurseries [37]. Additionally, we found that AM association was affected by high EC value and HM content (Pb, Ag, Mn, Zn, Fe, Sb, and Cu) as a consequence of mining. The Off-mine and the less disturbed areas (Exp 1 and Urban Ruins) showed the highest proportion of mycorrhizal plants and the highest levels of AM root colonization compared with the most degraded area (Exp 2). Particularly, *T. alata* plants showed high colonization rates in Exp 2, suggesting a remarkable tolerance and adaptation of this consortium to elevated HM levels in soils. Furthermore, plant species surveyed from the Boraginaceae, Malvaceae, Brassicaceae, and Geraniaceae families did not form AM symbiosis in the Paramillos de Uspallata mine and its surroundings. Although Lugo et al. [38] found that *Ehretia cortesia*, a member of the Boraginaceae family, established AM symbiosis in a dryland ecosystem; hence, further surveys of *P. sinuata* plants are required to determine its mycorrhizal status in this region. Mycorrhizal colonization may be influenced by many factors, including soil conditions (e.g., nutrient availability and soil degradation), microbial interactions, and plant traits [32]. For example, *S. philippiana* (Malvaceae family) has a high density of root hairs, which may reduce its mycorrhizal dependence. Similarly, plants from Brassicaceae, *E. cicutarium* (Geraniaceae), and *P. sinuata* (Appendix A) were heavily colonized by other fungal root endophytes, probably thus limiting the AM root colonization [32].

The present study highlighted the importance of considering mycorrhizal status as a driving force in ecosystem functioning, particularly in disturbed environments. Recent studies utilizing stable isotopes have demonstrated functional differences between Arum-type and Paris-type colonization. The Arum type enhances the mycorrhizal colonization length and facilitates nutrient transport to the host plant, whereas the Paris type is more effective in improving P acquisition and plant growth promotion [39,40]. In concordance with previous findings in arid habitats [41], most perennial plants surveyed in the Paramillos de Uspallata areas formed the Arum type, and less frequently, the Arum–Paris type in some individuals of *P. speciosa*, *J. uniflora*, *A. mendozana*, and *T. alata*, suggesting certain plasticity in mycorrhizal strategies. Our results suggest that plants forming Arum type in the mine may exhibit adaptations to low-nutrient conditions, typical of disturbed areas. Similarly, Matekwor et al. [42] proposed that the Arum type is associated with pioneer plant species, whereas the Paris type is more common in late successional stages of ecological successions and slow-growing plants [43]. The dominance of Arum type in *P. speciosa,* a pioneer and ruderal hemicryptophyte widespread across the mine and its surroundings, significantly contributed to the recolonization of disturbed environments, particularly in the highly polluted Exp 2 area within the mine.

Our proposed hypothesis was partially supported, as the analyzed AM fungal parameters (spore density, species richness, and diversity indexes) did not statistically differ between mine areas and the unexploited area. However, in the most contaminated area (Exp 2), we observed a low abundance and richness of plant species, coupled with a reduced percentage of AM root colonization. This was further associated with a decline in AM fungal richness, characterized by the dominance of an AM species with a high sporulation rate. These findings suggest that natural passive restoration in the most heavily impacted area may require more than 40 years to achieve ecological recovery.

By contrast, PCA results reveal clear differences among the study areas, with the first two axes explaining up to 86.9% of the total data variability. The variables that mostly contributed to this separation included the physicochemical soil parameters such as EC and TDS, as well as HM and other chemical elements. The Exp 1 area was mainly associated with AM spore density and concentrations of V, Ba, and Na, while Exp 2 was clearly separated and linked to elevated HM content (Pb, Ag, Mn, Zn, Fe, Sb, Cu), K and S, and high TDS and EC values in soils. Additionally, AM fungal richness showed an inverse relationship with Exp 2, highlighting the negative effect of soil pollution on fungal diversity in this highly disturbed environment. The Off-mine area was characterized by the highest AM fungal richness and Sr and Mg concentrations. Consistent with previous research in mines [25,44], the activities significantly exerted influence on these fungal parameters, particularly in the heavily impacted area (Exp 2). In contrast, the Urban Ruins area showed a strong association with P content, probably due to the accumulation of organic wastes from historical urbanization and elevated Ca ions that may precipitate P under alkaline conditions [45].

Conversely, the NMDS analysis highlighted the relationship between the most abundant species in each area and their association with environmental variables. While the results are not statistically conclusive, a clear trend emerged, with certain species demonstrating a notable capacity to thrive in the most adverse environments. The presence of species from the Acaulosporaceae family in soils with elevated levels of HMs is consistent with previous studies, which have documented the occurrence of various species in mining-impacted soils [46,47,48].

The most abundant AM fungal species were *E. infrequens* in Exp 2 and *E. etunicata* in Exp 1, Urban Ruins, and outside the mine. Likewise, *F. geosporus* (Glomeraceae) and *A. scrobiculata* (Acaulosporaceae) species were found in high proportion within the mine. All these AM species have been previously reported in the central Andes Mountains under extreme conditions [34], as well as in some disturbed or polluted environments [14,49,50,51]. In concordance with our results, Lugo et al. [52] found *Glomus sinuosum* associated with *P. speciosa* in a nearby location from Paramillos mine. A notable finding is the exclusive presence of certain species in the most disturbed area (Exp 2). Spores of *A. rehmii*, *S. constrictum*, and *S. baltica*, though found in low abundance, were detected solely in this area, suggesting the adaptability of these species to unfavorable environmental conditions. The dominance of *E. infrequens* in Exp 2 could be attributed to its dark-colored spores, associated with melanin content, which may enhance their survival under stressful conditions [29]. Additionally, other physiological mechanisms may also be involved, such as increased efflux and/or reduced HM uptake by hyphae, HM sequestration by metallothioneins, cell wall components, and glomalin, as documented in other AM species [6,7,53]. Consequently, these AM species that cope with harsh conditions within the mine could be considered stress-tolerant strategists.

*E. infrequens* exhibited the highest spore production in the most severely impacted area (Exp 2). This fungal trait was associated with elevated levels of EC, TDS, and HM contents. It is well established that sporulation can be influenced not only by the life history strategy of individual species but also by edaphic conditions (e.g., pH, nutrient availability), environmental factors (e.g., drought, soil compaction, altitude, latitude, depth), and the phenological state of the host plant [54,55]. The dbRDA analysis identified the EC parameter as a significant environmental variable positively associated with the presence of *E. infrequens* spores. Previous studies have described a positive correlation between spore production and high EC levels in various ecosystems, including the Chilean Matorral (a Mediterranean-type ecosystem), salt flats in the Atacama Desert, and arid and semi-arid Algerian regions [34,56,57,58,59]. This study contributes further evidence that EC may act as a trigger for AM sporulation, particularly under extreme environmental conditions. However, many of these propagules may be non-viable, which could explain the low level of AM colonization in root plants from Exp 2, although stress-tolerant species are characterized by their low intraradical mycelial development [51].

It is well established that the predominance of certain AM fungal life history strategies may vary across different successional stages and environmental conditions [18]. Additionally, AM symbiosis is widely recognized as a key driver of restoration through various mechanisms, including enhanced plant nutrition and mitigation of HM pollution [12,51]. As expected, ruderal species (primarily from the Glomeraceae family) prevailed in most mine sites, although stress-tolerant species became more dominant in the highly polluted Exp 2 area. The Urban Ruins and Exp 1 sites harbored AM strategists with proportions like those found in the Off-mine area, where ruderal species were dominant, followed by stress-tolerant species and, to a lesser extent, competitive species. This pattern suggests that natural passive restoration may be occurring in the Urban Ruins and Exp 1 areas due to a less severe mining impact, potentially representing a successional stage of advanced recovery. Likewise, the AM fungal species exhibit resilience and adaptability to disturbed environments [11,25,27], as well as the ability to disperse aerially and colonize these mining areas, particularly in this high-altitude arid environment with strong winds [60]. In contrast, the abundance of competitive strategists (e.g., Diversisporaceae), which thrive in undisturbed environments and require a high C demand and prolonged life cycles, increased in the Off-mine area.

These findings suggest that the dynamic shifts in AM fungal life history strategies during ecological restoration in mining areas may follow a predictable successional trajectory. This is consistent with previous studies that have reported similar trends in AM fungal community dynamics during ecological succession in other degraded environments [27,61]. Initially, stress-tolerant species prevail in severely degraded environments, withstanding extreme conditions caused by prolonged exposure to physical and chemical soil degradation. As edaphic conditions improve, ruderal species become more important, facilitating revegetation by rapidly re-establishing hyphal networks and symbiotic interactions. Eventually, competitive species increase in abundance in the ameliorated environment. In the unexploited area, which may represent a late successional stage of recovery, life history strategies were more balanced, indicating greater diversity within a stable environment. More studies are needed to confirm our findings and deepen our understanding of the natural restoration process in abandoned mines within arid environments, such as the Paramillos de Uspallata and other Andean mining sites.

Finally, our study highlights the importance of the Paramillos de Uspallata mine as a reservoir of stress-tolerant AM fungi, which have demonstrated successful function in bioremediation of polluted mine soils in experimental assays [21,22]. Further studies should focus on the potential of *P. speciosa* associated with *E. infrequens* as a consortium for ecological restoration and remediation of abandoned mines, particularly in highly impacted areas.

## 4. Materials and Methods

### 4.1. Description of Study Areas and Sampling

The Paramillos de Uspallata mine is an underground mine with a system of tunnels and galleries inside the mountains that cover a surface area of 120 km^2^. It is located on the western side of Sierra de Uspallata in the Andes Mountains (Mendoza province, Argentina, 32°29′10″ and 32°28′04″ S; 69°09′59″ and 69°08′19″ W, Figure 8a) with an altitude ranging from 2600 to 3100 m above sea level. The climate is arid with scarce precipitation that barely exceeds 200 mm^3^ per year, and the average temperature is 15 °C, ranging from −5 °C to 42 °C.

The study area corresponds to the phytogeographic region of Monte with Puna intrusion [62,63], which is characterized by sparse and low-cover vegetation, predominantly with xerophytic shrubs and a herbaceous steppe. The soil texture is sandy loam inside the mine (66.8% sand, 14.7% silt, 18.5% clay) and sandy clay loam (55.6% sand, 20.3% silt, 24.1% clay) in the area outside the mine, both with predominance of fine-grained matrix and large angular clasts (determined by Instituto de Geocronología y Geología Isotópica of Facultad de Ciencias Exactas y Naturales of Universidad de Buenos Aires). According to USDA classification, soils are classified as entisols and typical Torriorthents [64].

Sampling was carried out considering environmental heterogeneity, accessibility, and the mining zones defined by past exploitation activities. Four different areas were determined: (1) an urbanized area (0.02 ha), where the mine workers settled (referred to as ‘Urban Ruins’, Figure 8b); (2) an exploitation area (0.15 ha), where raw material grinding occurred (referred to as ‘Exp 1’, Figure 8c); (3) an exploitation and treatment area (0.12 ha) (referred to as ‘Exp 2’, Figure 8d,e), where extraction, processing, and storage of minerals took place, including treatment pools (Figure 8d) and a mine dump (Figure 8e); and (4) an undisturbed area outside the mine (0.15 ha) (referred to as ‘Off-mine’, Figure 8f). The distance between areas varied from 246.8 to 467.5 m, while the distance from the mine entrance to the Off-mine area was 3.4 km.

Four locations were selected within each sampling area, except for the Urban Ruins area, where three locations were chosen due to their smaller size compared with the other areas. Despite this, the number of samples was comparable to that in the other areas, ensuring that the sampling effort remained consistent. In each location, roots from at least three individuals of every plant species were collected to analyze mycorrhizal colonization. The number of samples collected was according to the number of plants found in each study location. Plants were taxonomically identified to the species level whenever possible.

Additionally, approximately 100 g of root-zone soil was collected to a depth of 15 cm from each plant and combined into a composite sample for the corresponding location. This resulted in four composite replicates per area. In the mine waste dump in the Exp 2 area, where no plants were present, three substrate samples were collected and combined into a single composite sample. All samples were sealed separately in polyethylene bags and stored at 4 °C until processed. Soil samples were air-dried and sieved through a 2 mm mesh for physicochemical characterization, HM quantification, and analysis of the AM fungal structures.

### 4.2. Soil and Substrate Analyses

Soil moisture (%H) was determined by drying 10 g of soil sample at 105 °C to a constant weight, while the TC content in samples from each area was estimated by incineration at 700 °C for 4 h. Also, the EC, TDS, and pH values were measured in a 1:2.5 suspension of soil–water using a pHmeter and conductivity meter (Hanna HI98130 model).

The total concentration of HM (Pb, Zn, Ag, Cu, Au, Cd, Ni, Cr, Sb, V) and other chemical elements (P, Ca, Mg, Na, K, S, Mn, Fe, Sr, Ba) in soil and substrate samples were determined using Wavelength Dispersive X-ray Fluorescence (WDXRF). For analysis, pressed pellets (28 mm in diameter) were prepared from 3 g of dried soil/substrate samples, and elemental concentrations were measured with a WDXRF S8 Tiger spectrometer (Bruker, Billerica, MA, USA) using the Quant Express (Standardless method) application at Laboratorio de Fluorescencia de Rayos X (Centro Atómico Constituyentes-CNEA, Buenos Aires, Argentina). Measurement control was performed using San Joaquin Soil (2709a) and Montana Soil (2710a) (NIST, Gaithersburg, MD, USA) as reference materials. The laboratory works under international standard for testing and calibration laboratories guidelines (ISO 17025). 

### 4.3. Biological Parameters

#### 4.3.1. Plant Traits

The plant traits of each species were determined according to the following criteria: functional type by using [65] classification [65] (hemicryptophyte, chamaephyte, succulent chamaephyte, therophyte or nanophanerophyte); life cycle (perennial, annual, or biennial) and the phenological state (vegetative, flowering, or fruiting). Additionally, the origin of each plant species (native, endemic, or exotic) was considered [66,67].

#### 4.3.2. Mycorrhizal Status

The mycorrhizal status and root colonization levels were assessed for all plants sampled within the mine and the unexploited area. Root samples were carefully washed under tap water to remove adhering soil particles and stained according to the modified Phillips and Hayman method [68]. Briefly, roots were cleared in 10% KOH solution at 90 °C for 20 min, washed with tap water, and acidified with 0.1 N HCl for 1 min at room temperature. Finally, roots were stained with Trypan blue in acid lactic (0.02%) at 90 °C for 15 min. Before the acidification, pigmented roots were bleached in 30% H_2_O_2_ solution for 1–2 min at room temperature. Fifty 1 cm root segments per plant were mounted on microscope slides with polyvinyl alcohol glycerol (PVLG) in groups of ten and examined using the Olympus microscope (model BX51) at 200× magnification.

The mycorrhizal type (AM association, ectomycorrhiza, ectendomycorrhiza, ericoid, or orchidoid mycorrhiza) and the AM colonization type (Paris, Arum, Arum–Paris) [69] were recorded for all sampled plants. For estimation of the intensity of AM root colonization of each sampled plant, three different levels of colonization were considered: low (<10% of root segments with AM structures), medium (>10% and <30%), and high (>30% of root segments with AM structures).

#### 4.3.3. AM Fungal Communities Analyses

In this study, morphological analyses of spores were used to characterize AM communities, given the increasing evidence that these traits are linked to life history strategies [29]. Briefly, AM fungal spores were extracted from 10 g of dry soil subsample, previously homogenized to eliminate typical heterogeneity of edaphic environment. The extraction process involved the wet-sieving and decanting method, followed by sucrose centrifugation [70,71]. The supernatant was passed through a 45 µm sieve, rinsed with tap water, and transferred to a Doncaster dish [72] for spore counting under a dissecting microscope at 40× magnification. Individual spores were carefully removed and mounted in Polyvinyl Lacto-Glycerol (PVLG) and PVLG + Melzer reagent (1:1) to examine their morphological characters using an Olympus BX51 compound microscope at varying magnifications. Whenever possible, identification at the species level was conducted based on morphological taxonomic criteria of spores (size, shape, color, ornamentation, subtending hyphae, and wall structures), following species descriptions [73,74]. Taxonomic assignments were performed according to the Index Fungorum (https://www.indexfungorum.org/names/names.asp (accessed on 13 February 2025). Permanent slides were deposited as reference material at the Banco de Glomeromycota in Vitro [75].

To determine the spore density in each area, the total number of spores was counted in 10 g of dry soil. The AM fungal composition was analyzed by calculating the relative abundance of each AM species as the proportion of spores of a species relative to the total number of spores in the soil sample. Species richness (S) values were estimated as the total number of AM fungal species per 10 g of dry soil for each area. The analysis of spore richness and abundance was conducted twice to ensure that all AM species were accurately recorded. In turn, trap cultures were established for each sampling location to detect any non-sporulating species presented at the sampling time.

Considering the prevailing soil conditions in each area, along with the multivariate analyses, the life strategies of each AM fungal species were inferred based on functional traits within Grime’s C-S-R model (competitor, stress tolerator, ruderal), following Chagnon et al. [18]. The proposal of a trait-based approach can contribute to our understanding of the ecological roles of mycorrhizal traits, successional dynamics, and the spatial structure of AM fungal communities, particularly in understudied environments such as the Paramillos de Uspallata mine.

### 4.4. Data Analysis

To assess the alpha diversity of AM fungi in the different areas, the spore density, species richness, and relative abundances were determined, and the Simpson and Shannon–Wiener indexes were calculated [76]. Differences in %H, TC, EC, TDS, spore density, and diversity parameters were analyzed using one-way ANOVA analysis after verifying the assumptions, followed by the post hoc Tukey test (alpha = 0.05).

A Bray–Curtis distance matrix was performed to quantify the dissimilarity in community composition based on the AM fungi data sets and to evaluate beta diversity across the different areas. Then, NMDS was applied to visualize differences, and a PERMANOVA test was performed to detect significant differences [77].

The relative contribution of soil parameters and biological data was estimated by PCA. In addition, dbRDA was performed to relate the relative abundance of each AM fungal species to the areas and significant environmental variables. These analyses were carried out using R software (version 4.3.2) with community analysis-specific packages, including BiodiversityR, Vegan, Devtools, and Factoextra (R Foundation for Statistical Computing 2022).

## 5. Conclusions

These results suggest that, after 40 years of passive restoration, some areas have exhibited natural self-recovery, depending on the prior mining activities. However, active interventions may still be needed to restore the most severely degraded areas, particularly through the use of local AM fungi and plant species. *E. infrequens*, predominantly associated with perennial and pioneer *P. speciosa* plants, demonstrated resilience or adaptation to elevated HM concentrations and high EC values, highlighting its life strategies as a stress-tolerant species. Further experiments are needed to confirm these findings and to assess the potential role of these AM species in post-mining restoration programs. However, replicating the complex environment of the mining ecosystem under controlled conditions is challenging, which makes this study valuable for shedding light on the life strategies of AM fungi.

## Figures and Tables

**Figure 1 plants-14-00580-f001:**
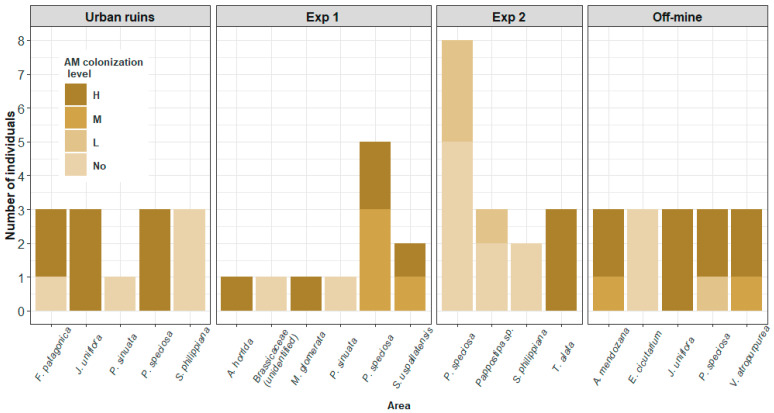
Number of plants of each species and level of AM root colonization in different areas within Paramillos de Uspallata mine and outside the mine (Off-mine). AM colonization level: high = H; medium = M; low = L; No: No AM root colonization. Urban Ruins: Urban Ruins area; Exp 1: Exploitation area 1; Exp 2: Exploitation area 2; Off-mine: Off-mine area.

**Figure 2 plants-14-00580-f002:**
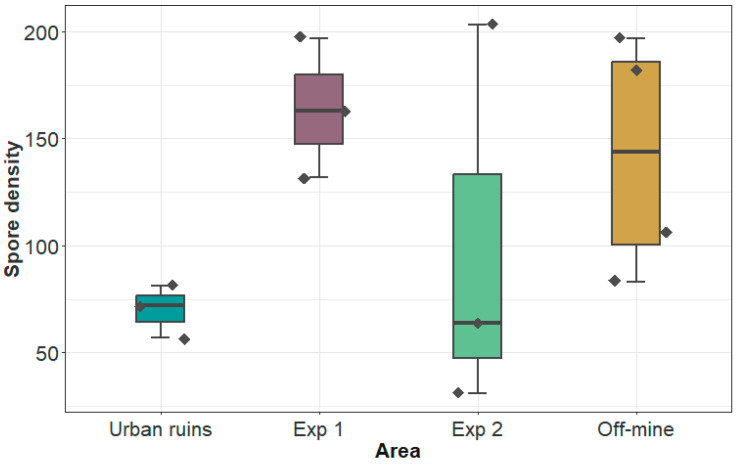
Spore density (number of AM spores per 10 g dry soil) for each area in Paramillos de Uspallata mine and an area without mining exploitation. The mean ± standard error values (bars) are reported. Urban Ruins: Urban Ruins area; Exp 1: Exploitation area 1; Exp 2: Exploitation area 2; Off-mine: Off-mine area. The ends of the whiskers represent the minima and maxima, the bottom and top of the box are the first and third quartiles, respectively, and the line inside the box is the median.

**Figure 3 plants-14-00580-f003:**
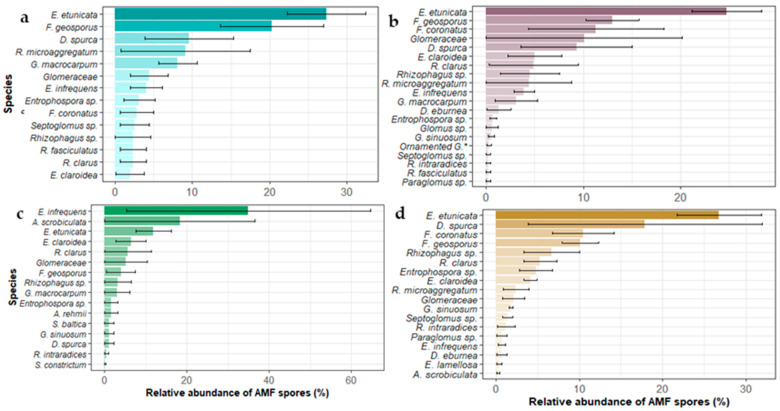
Relative abundance (%) of AM fungal species based on morphological taxonomy of spores for each area in Paramillos de Uspallata. The mean ± standard error values (bars) are reported. Urban Ruins (**a**); Exploitation area 1 (**b**); Exploitation area 2 (**c**); Off-mine area (**d**).

**Figure 4 plants-14-00580-f004:**
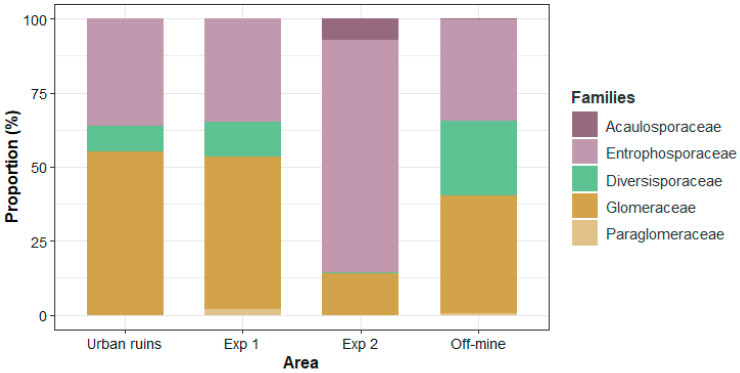
AM fungal families (%) found in each area. Urban Ruins: Urban Ruins area; Exp 1: Exploitation area 1; Exp 2: Exploitation area 2; Off-mine: Off-mine area.

**Figure 5 plants-14-00580-f005:**
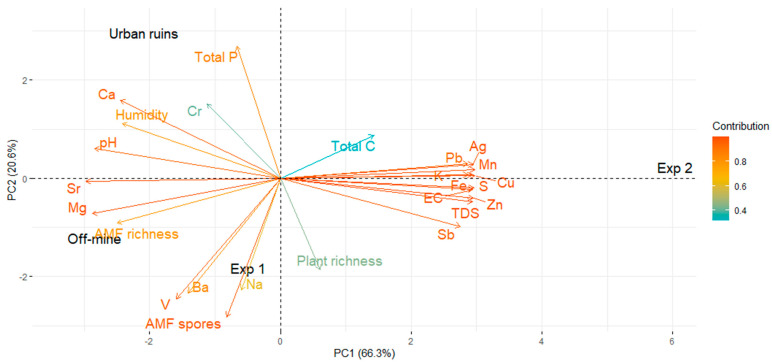
Principal component analysis for the environmental variables analyzed in each area within Paramillos de Uspallata mine (Urban Ruins; Exploitation area 1: Exp 1; Exploitation area 2: Exp 2) and outside mine (Off-mine area). The color code and the angles of the vectors indicate the contribution of each variable to the PC 1 and PC 2 axes. AMF spores: abundance of AM spores; AMF richness: richness of AM fungal species; TDS: total dissolved solutes; EC: electrical conductivity; Total C: total carbon; Total P: total phosphorus.

**Figure 6 plants-14-00580-f006:**
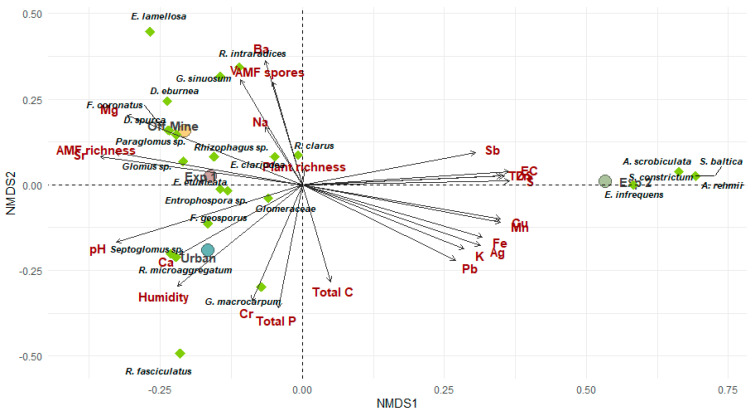
A non-metric multidimensional scaling (NMDS) plot performed using the Bray–Curtis dissimilarity matrix for the AM fungal communities and environmental parameters. Each green diamond represents AM fungal species and the color circles the areas of Urban Ruins; Exploitation area 1: Exp 1; Exploitation area 2: Exp 2, and Off-mine area. AMF spores: abundance of AM spores; AMF richness: richness of AM fungal species; TDS: total dissolved solutes; EC: electrical conductivity; Total C: total carbon; Total P: total phosphorus.

**Figure 7 plants-14-00580-f007:**
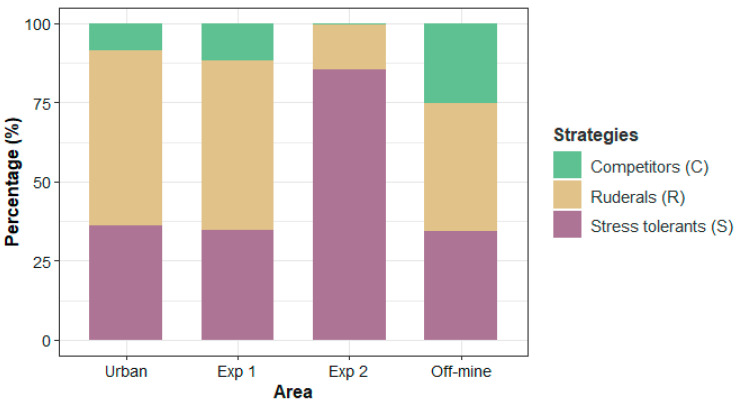
Proportionality (%) of AM functional groups based on the life history strategies C-S-R in each area within Paramillos de Uspallata mine (Urban Ruins; Exploitation area 1: Exp 1; Exploitation area 2: Exp 2) and outside the mine (Off-mine area).

**Figure 8 plants-14-00580-f008:**
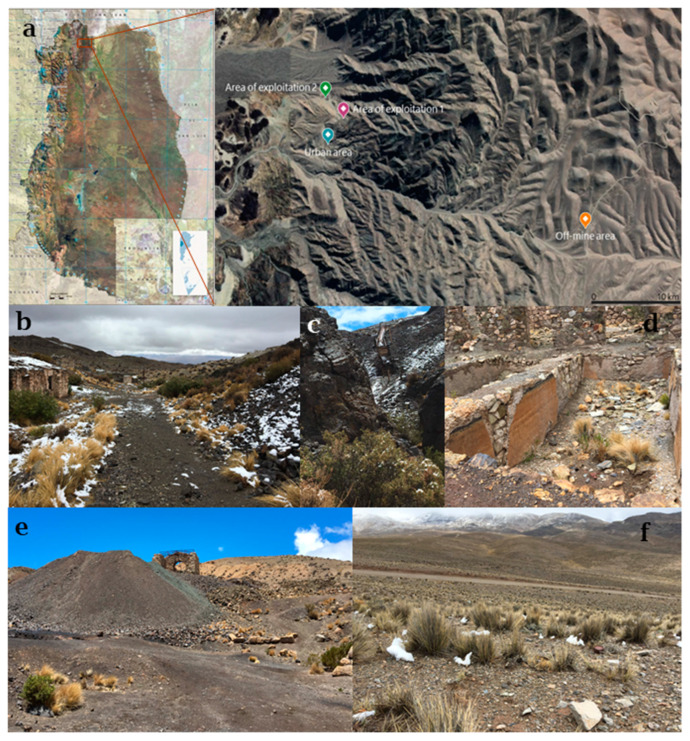
Geographical location of Paramillos de Uspallata mine (**a**) in Mendoza (Argentina) (**left**) and the sampling areas (**right**): Urban Ruins area (**b**), Exploitation 1 area (**c**), Exploitation 2 area (**d**,**e**), and Off-mine area (**f**).

**Table 1 plants-14-00580-t001:** Physicochemical properties and chemical elements concentrations of soils and substrate samples from Paramillos de Uspallata mine and an area outside of the mine.

Area	Urban Ruins	Exp 1	Exp 2	Exp 2-Dump	Off-Mine
**% H**	2.64 ± 0.14 a	2.57 ± 0.60 a	2.40 ± 0.10 a	2.33 ± 0.20 a	2.53 ± 0.19 a
**pH (H_2_O)**	8.4 ± 0.2 a	8.3 ± 0.1 a	7.8 ± 0.4 b	5.5 ± 0.1 c	8.3 ± 0.1 a
**EC (mS/cm)**	0.59 ± 0.1 a	1.33 ± 0.4 a	5.46 ± 2.5 b	3.57 ± 0.2 a,b	0.48 ± 0.2 a
**TDS (ppt)**	0.29 ± 0.1 a,b	0.67 ± 0.2 b	1.49 ± 0.4 c	1.78 ± 0.1 c	0.21 ± 0.1 a
**TC (%)**	7.18 ± 0.47 a	7.31 ± 0.60 a	7.10 ± 0.25 a	6.69 ± 0.42 a	5.47 ± 0.79 a
**TP (%)**	**0.43 ± 0.04**	0.18 ± 0.02	0.16 ± 0.02	0.12 ± 0.01	0.12 ± 0.01
**Ca (%)**	**6.20 ± 0.30**	3.20 ± 0.20	2.10 ± 0.10	2.40 ± 0.10	5.40 ± 0.30
**Mg (%)**	1.37 ± 0.07	1.52 ± 0.08	0.78 ± 0.08	0.35 ± 0.04	**1.65 ± 0.08**
**Na (%)**	0.81 ± 0.08	**1.10 ± 0.06**	0.97 ± 0.07	0.27 ± 0.03	0.89 ± 0.09
**K (%)**	2.40 ± 0.10	2.45 ± 0.10	**2.60 ± 0.10**	2.00 ± 0.10	2.30 ± 0.10
**S (%)**	0.31 ± 0.03	0.71 ± 0.07	2.80 ± 0.10	**8.40 ± 0.40**	0.15 ± 0.02
**Mn (%)**	0.46 ± 0.05	0.53 ± 0.05	1.85 ± 0.12	**2.00 ± 0.10**	0.11 ± 0.10
**Fe (%)**	5.20 ± 0.30	5.70 ± 0.30	**8.15 ± 0.30**	**7.30 ± 0.40**	4.20 ± 0.20
**Zn (%)**	0.16 ± 0.02	0.79 ± 0.08	**2.47 ± 0.08**	**2.80 ± 0.10**	0.020 ± 0.002
**Pb (%)**	0.43 ± 0.04	0.53 ± 0.05	**1.17 ± 0.75**	**1.00 ± 0.10**	Det < 0.005
**Cu (mg Kg** **^−1^)**	182 ± 18	248 ± 25	**726 ± 68**	239 ± 24	51 ± 10
**Ag (mg Kg** **^−1^)**	101 ± 14	117 ± 15	**339 ± 34**	**388 ± 39**	ND < 50
**Cd (mg Kg** **^−1^)**	ND < 50	ND < 50	**Det < 100**	**Det < 100**	ND < 50
**Cr (mg Kg** **^−1^)**	**70 ± 14**	**64 ± 13**	ND < 50	ND < 50	ND < 50
**Au (mg Kg** **^−1^)**	ND < 50	ND < 50	ND < 50	ND < 50	ND < 50
**Sr (mg Kg** **^−1^)**	267 ± 27	253 ± 26	106 ± 15	81 ± 16	**301 ± 30**
**Ba (mg Kg** **^−1^)**	446 ± 45	550 ± 55	483 ± 49	**1300 ± 100**	622 ± 62
**Ni (mg Kg** **^−1^)**	**57 ± 11**	49 ± 10	39 ± 8	48 ± 8	32 ± 6
**Sb (mg Kg** **^−1^)**	ND < 50	180 ± 18	187 ± 19	**214 ± 21**	ND < 50
**V (mg Kg** **^−1^)**	ND < 50	**173 ± 26**	ND < 50	ND < 50	138 ± 21

%H: percentage of soil moisture; TC: total carbon; TP: total phosphorus; EC: electric conductivity; TDS: total dissolved solutes. For %H, TC, EC, TDS, values represented the average ± standard error. Different letters within areas indicate significant differences according to the Tukey test (alpha = 0.05). For chemical elements measured by WDXRF, the reported values represent the total concentration of each element ± the combined standard uncertainty per area. The maximum concentration for chemical elements is indicated in bold. ND: no detected value, below the detection limit, Det: detected value. Exp 1: Exploitation area 1; Exp 2: Exploitation area 2; Off-mine: Off-mine area.

**Table 2 plants-14-00580-t002:** Family and plant species (with origin in brackets), plant traits, and mycorrhizal status in each study area within Paramillos de Uspallata mine and an area located outside the mine.

Family/Species of Plants	Area (Number of Plants)	Plant Traits			Mycorrhizal Colonization Type
		Functional Type	Life Cycle	Phenology State	
Poaceae *Pappostipa speciosa* (n)	Urban (3) Exp 1 (5) Exp 2 (8) Off-mine (3)	H	PH	v	A-P, A
*Pappostipa sp.* (n)	Exp 2 (3)	H	PH	fl, v	A
Boraginaceae *Phacelia sinuata* (en)	Urban (1) Exp 1 (1)	C	PH	fl	-
Solanaceae *Fabiana patagonica* (n)	Urban (3)	N	PS	v	A
Verbenaceae *Junellia uniflora* (en)	Urban (3) Off-mine (3)	C	PSS	fl	A-P, A
Malvaceae *Sphaeralcea philippiana* (en)	Urban (3) Exp 2 (2)	C	PH	fl v	-
Asteraceae *Senecio uspallatensis* (en *) *Artemisia mendozana* var. *paramilloensis* (en *)	Exp 1 (2) Off-mine (3)	N N	PS PSS	fl v	A A-P, A
Brassicaceae *Unidentified species* (ex)	Exp 1 (1)	T	AH	v	-
Fabaceae *Adesmia horrida* (n)	Exp 1 (1)	N	PS	fl	A
Cactaceae *Maihueniopsis glomerata* (n)	Exp 1 (1)	SC	PSS	fl	A
Rosaceae *Tetraglochin alata* (n)	Exp 2 (3)	C	PSS	v	A-P
Geraniaceae *Erodium cicutarium* (ex)	Off-mine (3)	T	AH	fl	-
Violaceae *Viola atropurpurea* (en)	Off-mine (3)	H	PH	v	A

Urban: Urban Ruins area; Exp 1: Exploitation area 1; Exp 2: Exploitation area 2; Off-mine: Off-mine area. Functional type: H = hemicryptophyte, C = chamaephyte, SC = succulent chamaephyte, T = therophyte, N = nanophanerophyte. Life cycle: PH = perennial Herb, PS = perennial shrub, PSS = perennial subshrub, AH = annual herb. Origin: n = native, en = endemic, e = exotic. Phenological state: v = vegetative, fl = lowering. Mycorrhizal types: AM = arbuscular mycorrhiza. No detection of Mycorrhiza. AM colonization type: A = Arum, P = Paris, A-P = Arum–Paris. * indicates endangered plant species.

**Table 3 plants-14-00580-t003:** Richness values (S) and Shannon (H) and Simpson (D) indices based on morphological characterization of AM spores for each area in the Paramillos de Uspallata mine and the unexploited area.

Area	S Value	H Index	D Index
Urban Ruins	10.67 ± 1.86	1.90 ± 0.22	0.18 ± 0.03
Exp 1	12.67 ± 2.02	1.98 ± 0.13	0.17 ± 0.02
Exp 2	7.33 ± 1.45	1.28 ± 0.54	0.44 ± 0.23
Off-mine	11.25 ± 0.75	1.84 ± 0.18	0.24 ± 0.07

Mean ± standard error values are reported. Urban Ruins: Urban Ruins area; Exp 1: Exploitation area 1; Exp 2: Exploitation area 2; Off-mine: Off-mine area. No statistically significant differences were found among areas (*p* > 0.05).

## Data Availability

All data are contained within the article.

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
