# Peer review of "Responses of Arbuscular Mycorrhizal Fungi and Plant Communities to Long-Term Mining and Passive Restoration"

_plants, 2025, doi:10.3390/plants14040580_

Round 1

Reviewer 1 Report

Comments and Suggestions for Authors

In the manuscript(plants-3463098) entitled ‘Responses of arbuscular mycorrhizal fungi and plant communities to long-term mining and passive restoration’ by Sofía Yasmín Utge Perri et al.. The study focuses on the Paramillos de Uspallata mine in the Andes Mountains and explores the responses of arbuscular mycorrhizal fungi (AM fungi) and plant communities during long-term mining and passive restoration. The topic has important ecological significance. The study analyzes the correlations among soil properties, plants and mycorrhizal status, and AM fungal communities in mining areas and unmined areas, and discusses the life history strategies of AM fungi, providing valuable references for ecological restoration in mining areas. However, I have some comments that should be addressed by the Authors prior to the publication.

1. Introduction, The introduction of the general functions of AM fungi in ecosystems is relatively detailed, but there is a lack of a more targeted review of the research status in South America, especially in mining areas of Argentina. Some specific cases and data of AM fungi research in mining areas in this region or under similar ecological conditions should be added to better highlight the uniqueness and necessity of this study.

2. Introduction, The research hypothesis is clear and reasonable. However, when elaborating on the hypothesis, relevant theories or previous research results can be further cited to enhance its logic and persuasiveness. For example, when referring to the life history strategy of AM fungi, the application and expectation of Grime's CSR model in similar studies can be described in detail.

3. Result, Soil properties: When presenting soil physical and chemical properties and chemical element concentration data (Table 1), significant notations of differences in various parameters between different regions (such as using different letters to represent) can be added to make it easier for readers to intuitively see whether the differences are significant. At the same time, for some special changes in soil properties, such as the potential ecological impacts of acidic pH values and high sulfur content in the Exp 2 region, a brief discussion should be carried out in the text.

4. Result, Plant and mycorrhizal status: When describing the plant species composition and mycorrhizal infection status (Table 2 and related paragraphs), for plant families that do not form mycorrhizal symbiosis, further explore the possible ecological or physiological reasons instead of just mentioning existing studies. In addition, for the mycorrhizal infection status of endangered plant species Senecio uspallatensis and Artemisia mendozana var. paramilloensis, emphasize their potential significance in conservation biology and discuss in combination with relevant conservation strategies.

5. AM fungal community: When analyzing AM fungal spore density, species composition and diversity index, although statistical tests have been conducted, for some results that did not reach significant differences (such as the comparison of spore density between different regions), possible reasons should be further analyzed, such as whether the sample size is sufficient and the influence of soil heterogeneity. When describing the changes in the relative abundance of AM fungal species (Figure 3), some brief descriptions of the ecological characteristics of dominant species can be added to help readers better understand their adaptability in different environments.

6. Discussion, In the discussion section, the analysis of some important findings can be more in-depth. For example, when explaining the dominance of Entrophospora infrequens in the Exp 2 area, in addition to correlating environmental factors, possible physiological and molecular mechanisms such as tolerance to heavy metals and absorption and transport mechanisms can be further explored. For the dynamic changes of different AM fungal life history strategies in the process of ecological restoration in mining areas, a more detailed model construction or prediction can be carried out by combining ecological succession theory.

7. M&M, In the analysis methods of soil and biological parameters, for some key steps, such as the specific operation parameters and quality control measures of determining chemical element concentrations by X-Ray Fluorescence technology, more detailed explanations should be provided to ensure the repeatability of the research. In the process of extracting and identifying AM fungal spores, for some species that are difficult to identify, it should be explained whether other molecular biology methods are subsequently adopted for auxiliary identification and the reasons for not being adopted.

Reviewer 2 Report

Comments and Suggestions for Authors

The manuscript submitted by Utge Perri and colleagues describes the arbuscular fungal communities found in a former mining area with different degrees of contamination. The presence of AM fungi was correlated with soil parameters. The highly contaminated area showed an enrichment of Entrophosporaceae and a depletion of Diversisporaceae compared to the other experimental sites. The authors conclude that initially, stress-tolerant AM fungi dominate in highly contaminated areas and are later, when conditions improve, replaced by other species and an overall more even distribution of different of AM fungi can be found.

The study is interesting and well done, but limited in scope as only four experimental sites were analyzed, and the number of plant individuals is low (mostly 3), while conclusions drawn go in part very far, in particular in lines 437-449, but also in other parts of the manuscript. This is a single experimental site where this distribution of AM fungi was observed, and though the reasoning is conclusive it cannot be ruled out that other factors not analyzed in this study may play a key role. It could be better highlighted that the conclusions drawn are assumptions but further studies are needed to test these hypotheses, in particular experimental and not only descriptive studies.

I do not rule out that I missed the information but I could not find how the fungi were classified in the different life-history strategies. Please provide more details on how the classification was done.

Minor comments:

line 190 relative species abundance

line 242 was observed

line 374 delete by contrast

Author Response

We uploaded our responses as a PDF file.

Round 2

Reviewer 1 Report

Comments and Suggestions for Authors

I would like to thank the authors for their detailed responses to the questions I asked and I agree with the authors' responses. I have no further comments on the latest revised manuscript.